# Emerging Stakeholder Relations in Participatory ICT Design: Renegotiating the Boundaries of Sociotechnical Innovation in Forest Biosecurity Surveillance

**Andrea Grant [1,*], Stephen M. Pawson [1] and Mariella Marzano [2]**

[1] Scion, 10 Kyle Street, Riccarton, Christchurch 8011, New Zealand; steve.pawson@scionresearch.com
[2] Forest Research, Northern Research Station, Roslin, Midlothian EH25 9SY, UK; mariella.marzano@forestresearch.gov.uk
* Correspondence: andrea.grant@scionresearch.com

**Abstract:** Research Highlights: This research advanced understanding of stakeholder relations within the context of innovation using citizen science in a biosecurity sociotechnical system (STS) in Aotearoa, New Zealand. Background and Objectives: It draws on recent experiences in the United Kingdom, where analysis of stakeholder engagement in the development of biosecurity surveillance technologies and citizen science initiatives have occurred to support understanding and development of forest and tree health biosecurity. Early detection technologies are essential as biosecurity risks to the primary sectors increase with the expansion of global trade and shifting pest dynamics that accompany a changing climate. Stakeholder engagement in technology development improves the chances of adoption but can also challenge the mental models of users in an existing STS. Materials and Methods: Two conceptual models that embed stakeholder relations in new information and communications technology (ICT) design and development were applied: (i) a future realist view of the general surveillance system incorporating citizen experts as species identifiers; (ii) a social construction of the ICT platform to surface mental models of the system in use creating the groundwork for evolution of stakeholder relations within STS innovation. A case study demonstrating how we addressed some of the practical limitations of a proposed systems change by applying sociotechnical innovation systems (STIS) theory to the development and adoption of new technologies for surveillance in the existing biosecurity system was presented. Results: Opportunities to enhance the capacity for early detection were considered, where the needs of diverse factors within a central government biosecurity authority and the wider citizenry are supported by the development of a general surveillance network (GSN).

**Keywords:** New Zealand; biosecurity; surveillance; invasive species; early detection; sociotechnical innovation; systems change; methodological pluralism

## 1. Introduction

### 1.1. Citizen-Enhanced General Surveillance

An enhanced general surveillance network (GSN) was proposed by Scion and Lincoln University research entomologists in Aotearoa, New Zealand, to provide a platform for the greater public (citizen) and commercial (industry) participation in plant biosecurity. Improved technology, e.g., mobile phone applications, they proposed, would enable better information exchange between GSN participants and would likely increase the effectiveness of public contributions to biosecurity surveillance. The speed and quality of the image and accompanying data submitted relative to current descriptive records given

in personal phone calls were expected to improve. At the same time, they proposed, such a network could increase the pest identification abilities of data submitters. Subsequently, a research programme was designed to bring together technology change with enhanced engagement to expand the roles of citizens as submitters and as experts in the identification of pest (including insects, pest plants, and pathogens that are visually distinguishable) observations to support central and regional government surveillance activities. The researchers anticipated that a communication network could address the shared needs of participants not only from the forestry sector but across different stakeholder groups within key central and local government agencies, other primary industry sectors and local and indigenous communities (Iwi is the traditional ancestral tribes of Māori, the indigenous people of Aotearoa, New Zealand). Furthermore, if successful, the network could incorporate real-time analytics to deliver spatially explicit alerts and provide feedback to individuals on observations made. Such improvements would increase the overall capacity of the existing biosecurity surveillance system, and enhance learning experiences of technology users, extending across a range of biosecurity settings. This paper addresses the challenges that such a transformation of general surveillance information systems would meet by altering the relationships between stakeholders in the existing sociotechnical system (STS).

*1.2. Applied Social Research in Biosecurity Technologies*

Social research in the UK has facilitated discussion within interdisciplinary teams on how social understanding needs to be embedded in biosecurity surveillance technology innovation, for example, through citizen scientists' involvement in tree health monitoring [1–3]. Further research has acknowledged that the public costs of biological invasions, increasing needs for budget austerity and a growing risk of pest incursions from trade and changes in climate warrant closer engagement of stakeholders, including citizens in the active aspects of biosecurity surveillance [3,4]. However, there have been limitations noted in the development of stakeholder and community engagement as they apply to biosecurity technology development for tree health, with only marginal potential gains for STSs transformation [2,5]. In New Zealand, additional issues of social and cultural acceptability of management technologies have become increasingly important in forest biosecurity risk [6–8]. Managing complex interactions between social and technical aspects of biosecurity has been a challenge for planted, exotic, urban and native forests and trees [9–13]. In some instances, local knowledge has been important to the development and application of invasive species mapping technologies [14], especially in the developing world where species distribution mapping has been limited. Further interest in mobilising higher levels of engagement in invasive species management or local eradication has also occurred in other parts of the developed world by including decision-makers in research [15]. Each setting has shown context to be important, and careful attention needs to be paid to the willingness to participate in biosecurity operations, including aspects of the enjoyability of the experience and interaction, as well as levels of time commitment and resources needed, and variations in usability of technology suitable for different needs [2,3,16].

In the UK, the growing interest in citizen science projects has seen the landscape and nursery sector playing an increasingly important role in supplementing the government capacities to monitor tree health [2]. In New Zealand, government agencies have had a statutory responsibility to manage biosecurity surveillance since 1993, including obligations for private property inspection (Biosecurity Act, 1993) [17], with similar regulatory tools emerging in the UK more recently (Infrastructure Act, 2014) [18]. Opportunities for the public and primary sectors to submit general surveillance observations in New Zealand is limited to a pest and disease hotline run by the central government. However, recently instituted Government Industry Agreements (GIAs) on Readiness and Response for biosecurity support shared governance and resourcing of primary industries biosecurity activities related to preparedness for biosecurity incursion response operations [19]. These agreements include a commitment to maintain capacity in biosecurity surveillance. To date different primary sector signatories have taken an ad-hoc approach to this with no specific pan-sector surveillance initiatives.

### 1.3. Designing New Technologies with Stakeholders

When considering how to design technologies, there are challenges that not only deal with technical problems—such as whether something works effective—but problems of social acceptability [20]. Developing or improving systems can involve a complex set of relationships and require the dismantling of existing systems, or renegotiation of new systems or ways of doing things. The way a research project or system improvement is defined and developed evolves out of the relationships between the researchers, funding bodies and research stakeholders [21,22]. Through such relationships, cultural contexts of design can constrain how user inputs are sought and included in technological innovation or influence why a particular direction in technology design is considered important [23]. Furthermore, project management can conflict with the emergent conditions of new technology, when governed by short timelines for deliverables that leave little room for negotiated outcomes of systems changes. Thus, there is a need for more artful approaches to integrating users in design processes [24]. In a research and innovation activity involving diverse groups of people working with different knowledge systems, it is important to know when to take stock and reconsider the nature of relationships, potential changes in roles or responsibilities and how that can affect the success of the impending improvement [24]. Initially, change can lead to frustration and additional burdens in work patterns requiring new cognitive routines [25]. Adaptation to new methods or processes requires some reduction in efficiency at first to learn new operations and become accustomed to using new technologies or innovations [26]. Less attention is paid to changed relationships within operational environments [27].

In this paper, we outlined a two-step process of conceptualising and reconceptualising the development of stakeholder relations in a GSN, as a proposed improvement to New Zealand's biosecurity system—adopting new mobile information and communications technology (ICT)—was added to the mix. We first described the problem setting involving New Zealand's biosecurity system, the challenges of maintaining border protections under increasing levels of invasive species risk and the opportunities for developing enhanced early detection through the engagement of citizens as data submitters as well as species identifiers. We then outlined our research aims and methodology using action research to support stakeholder engagement in the design and development of STS change. Our results were presented, and the outcomes of our applied methods of stakeholder engagement were described with consideration of the challenges of socially constructing a GSN through diverse perspectives. We concluded with a project reconceptualisation and new set of challenges for the development of stakeholder relations in the evolution of using new mobile ICTs in the biosecurity general surveillance system. This highlights the importance of acknowledging the deliberations of diverse actors in different parts of the biosecurity system to overcome institutional barriers to systems innovation.

## 2. Background

### 2.1. New Zealand's Biosecurity System

New Zealand's biosecurity system is layered within the broader overarching structures of the World Trade Organisation's (WTO) Sanitary and Phytosanitary measures [28], which New Zealand interprets through the lens of its trade agreements and bilateral arrangements (e.g., Free Trade Agreements). This means that the global standards for risk assessment and biosecurity protocols for plant and animal health governing incursion response are embedded within the conditions for market access. As per relevant International Plant Protection Convention standards, the biosecurity risks of trade are assessed using pest risk assessments [29] and inform New Zealand's Import Health Standards. These standards are designed to protect New Zealand from biosecurity threats, and compliance is checked via inspections at the border as well as pre-border movement protocols. Providing confidence to trade partners is a key element of New Zealand's biosecurity and surveillance system. However, the risk of incursions through trade or other human-directed and naturally occurring movements can never be reduced to zero. New Zealand invests significantly in the minimisation of biosecurity risks through

border protection and pre-border movement protocols and inspection activities, yet new and emerging threats and ways to treat them remain a concern [30]. As part of targeted post-border surveillance, thirteen surveillance programmes are operated by Biosecurity New Zealand (a government agency), often termed 'active' surveillance, to monitor for high-risk pests, at high-risk locations, and of vulnerable groups of plants and animals in both terrestrial and aquatic ecosystems [31]. These programmes are managed to maximise early detection probabilities for new incursions and to facilitate trade into overseas markets by confirming New Zealand's pest- and disease-free status. Early detection, although notoriously difficult, is crucial as it is one of the best predictors of eradication success and minimises eradication costs [32,33].

Developing a stronger surveillance system for early detection is one way of reducing uncertainty and improving opportunities for successful eradication should an incursion occur. Early detection is key to successful eradication, and existing general surveillance programmes for forest pests often fail to detect species at a stage earlier enough to facilitate eradication [9]. Improved early detection can also provide information for the biosecurity system specific to New Zealand, as it adapts to new challenges that may present now and in the future, that may have relevance for other countries signed up to the WTO [34]. These include increased trade and movement of people [35,36] as well as uncertainty around the environmental conditions of pest entry, establishment and spread, especially under climate change [30,34,37].

### 2.2. Enhancing Biosecurity Surveillance

Biosecurity New Zealand (BNZ) and its associated networks (e.g., AsureQuality—contracted government corporation servicing biosecurity operations; Regional Councils—a second tier of government between national and local authorities; industry associations—such as Forest Owners Association, Apples and Pears New Zealand and Kiwifruit Vine Health) are the main actors involved in biosecurity surveillance practices. Further capacity exists within communities to contribute to general biosecurity surveillance through the data submission or species identification of unusual pest sightings. However, community resources are not as engaged as they could be [38]. Community participation in reporting a suspicious or new-to-New Zealand pest or disease currently relies on a '0800' free phone number. The New Zealand government's 10-year biosecurity strategic direction statement seeks greater public participation in biosecurity, including surveillance through the '4.7 Million Eyes' aspirational goal of bringing the whole of the New Zealand populous into biosecurity surveillance activities [39]. Yet additional reporting will increase pressure on existing government biosecurity surveillance and investigation resources, under stress from an increasing number of incursions investigations and responses activities, e.g., *Bonamia ostreae* Pichot (a disease of oysters), *Microplasma bovis* Karlson & Lessel (a disease of cattle), Queensland fruit fly (*Bactrocera tryoni* Froggatt, a horticultural pest) and myrtle rust (*Puccinia psidii* Winter, a disease impacting Myrtaceae species, including exotic and native plants of conservation, production and cultural value), as the more high profile examples within a two year period (2017–2019) (See Biosecurity New Zealand's Surveillance magazine for a more extensive log of incursion investigations (BNZ, 2019)). A step-change in operations is required to support greater community participation in general surveillance [40] and make use of available ICT capabilities, such as smartphone applications. There is an identified need for understanding how existing reporting channels could handle more public submissions, including clearer and more standardised information collection [40]. Other areas of potential improvements in social and technical aspects of surveillance include appropriate and well-functioning networks and timely and accurate notifications [41].

Marzano et al. [2] noted the importance of increasing the adoption of new technologies for tree health in a rapidly changing world as trade amplifies the risk of spreading invasive species. However, they also highlighted challenges for those developing new technology, e.g., that may have limited markets and, therefore, a limited avenue for adoption. Our challenges are less related to the smaller market pools as New Zealand has a national strategy to engage the entire population in biosecurity activities. The challenge is one of adoption, not of something novel but of the citizen

science contribution that disrupts the existing model of biosecurity surveillance. For New Zealand, the bigger challenge relates to the change in systems of operation and the implications of sociotechnical innovation [27] on those currently managing the biosecurity system. We report on not a science-based innovation but a social and technological innovation that draws on greater external capacities of New Zealand citizens and industries to support the existing government surveillance system for early detection of biological invasions. Further to the dynamics of disruption, "tree health policy [in the UK] is responding rapidly to social change (such as globalisation and trade) and environmental change (such as climate change and alien invasive species), shifting the context and needs for detection technologies" [2] (p.28). New Zealand is similarly exposed to these landscape-level changes, in which biosecurity regimes are also confronting the changing environment of social and cultural acceptability of biosecurity technologies and operations [6,7,20]. This adds pressure to innovate in the space of early detection.

Research Aims

This paper focused on bringing stakeholders in general biosecurity surveillance from New Zealand around the table to discuss biosecurity issues and in the process contribute towards a sociotechnical innovation that could support the government aim of improving the efficiency of the general surveillance system. Our research aimed to involve Iwi, sector and agency stakeholders in the development of design criteria for the new biosecurity system tools, specifically mobile phone applications to interface with other information systems, such as iNaturalist NZ (iNaturalistNZ is based within the platform (https://www.inaturalist.org/) from the US as a networked educational initiative where community-based skill and expertise is used to submit and identify images of plants, animals and fungi taken by public contributors to the initiative). To support this, we needed to understand and map the evolving dynamics of stakeholder engagement in the researchers' proposed vision of a GSN. The research team comprised biological and social scientists with skills and capabilities in biosecurity, entomology, research management, sociology, systems thinking and Māori research methods (kaupapa Māori). We engaged and included the contribution of biosecurity managers and practitioners to develop their capacity to further engage a broader group of stakeholders from their networks. The objective was to move stakeholder engagement further 'upstream' in the technology development process [42,43] to support the inclusion of technology end-users when developing the ICT innovation (such as a mobile app and its associated information processing systems) design and to build on existing stakeholder networks. We adopted an approach that aimed to work through existing networks of representative stakeholders (Iwi/agency/sector organisations) to connect the ICT innovation with their wider communities. The approach adopted was premised on bi-directional knowledge exchange through a networked community, rather than one that worked through conduit and container models of technology transfer, that is usually attempted at the end of technology development and inadequately addresses the knowledge and frameworks of users [44].

## 3. Materials and Methods

A co-design approach using the process of stakeholder engagement to guide the design and development [45,46] of a biosecurity STS innovation was followed. We called this innovation a *surveillance platform* comprising an app and a potential network of users. Action research, an essential element of co-design [47], ought not to simply support the intent of stakeholders but to introduce social science questions to challenge or reflect on the bias of underlying assumptions that may practically or ethically impact the anticipated outcomes [48]. Qualitative research methods were used to facilitate the development of an action research framework and a set of questions to incorporate stakeholder considerations into technology design processes, for engaging potential technology users in a different sector, agency and iwi communities. A key element of our action research was to try and capture stakeholders' mental models of biosecurity surveillance or what that system looked like in the minds of stakeholders and how that would be enhanced by our ICT innovation. Mental models are the ideas

held within the minds of a process or technology user of what the system they are using looks like [49]. This includes the way a person interprets and interacts with a situation, the way they make sense of the things that are happening and how they incorporate them within their activities and responses [50]. Participatory approaches to biosecurity technology design were adopted where researchers worked with key stakeholders early in the research process to develop a set of questions and a framework for the engagement of potential users in the ICT innovation [51]. Stakeholders from plant and animal industries, primary industries (including forestry) and conservation agencies, regional government and an iwi-based organisation were invited to participate in a technology co-development workshop and to collect further data through engagement with their stakeholder networks, initiating the development of an enhanced GSN for biosecurity.

### 3.1. Processes Developed

The focus of the engagement process was on the front-end of technology development (i.e., the user interface of the mobile tool) rather than the back-end of receiving data submissions within the existing biosecurity surveillance system. Methodologically, there were three main aspects to consider in developing relationships with key stakeholders and asking them to engage further stakeholders through their networks:

- Building skills for engaging stakeholders
- Offering practical experience of research
- Connecting through existing networks.

Some stakeholders were leaders in this initiative and had invested in the experiment to test the viability of the concept. Challenges were acknowledged in being able to get people to use yet another app in a crowded market place; and having the support of the main protagonist, the government agency responsible for incursion investigations. The New Zealand Forest Owners Association was one of the key investors and others committing resources to the project included, Biosecurity New Zealand, Regional Councils and Kiwi Vine Health. Dairy NZ, Beef and Lamb NZ, Wine Growers New Zealand, HortNZ and Avocados NZ remained observers of the tool development, with potential to invest in the joint ownership of the platform at a future date.

### 3.2. Data Collection Methods

In the design of the research, three main methods were used to collect and incorporate data into our learning processes: meetings; workshop; surveys (Table 1). Meetings were held both during the proposal development and before and following the design workshop. They were a critical part of the research design process and included the capture of design elements, risk management and willingness to engage. An iterative process was used to capture information from meeting participants and send summaries back for reflection and feedback on the directions being taken. The workshop allowed for further engagement in technology design, including appreciating technology success/failures, determining technology specifications (function, form and content), considering minimal data requirements, and mapping stakeholder networks. Participants were recruited from various industries, agencies, and additional meetings were held with an iwi organisation to support the development of design suitable for Māori uses. A small co-design team was nominated by the workshop participants to help develop the functionality of the tool as well as the design of a stakeholder survey. A survey was conducted by workshop participants of people who would potentially use the app and platform from participants' networks to find out about current surveillance practices, technologies in use, trusted advisers and what kind of feedback they'd like to receive. The number and type of people involved in each of the three data collection methods are shown in Table 2.

**Table 1.** Mixed methods of data collection according to their purpose, process and outcomes.

| Data Collection Method | Purpose of Method | Process Followed | Outcomes Provided |
|---|---|---|---|
| Meetings | early engagement, diverse perspectives | iterative discussion, review, feedback | critical elements, agreed boundaries, managing expectations, mitigating risks |
| Workshop | co-design aspects, define success, understand improvements | discuss style and minimum requirements, understand potential use, scope function and form | minimum data requirements, stakeholder environments/networks mapped, co-design team formed |
| Survey | broader engagement, exploring benefits, extending networks | questions co-designed, guided conversations, participant implemented | pest detection reactions, technologies in use, information shared, desired feedback, reporting constraints |

**Table 2.** Data collection methods, who participated and how many people were involved.

| Data Collection Method | Type of Participant | Number of Participants |
|---|---|---|
| Meetings | Sectors | 4 |
| | Ministries | 2 |
| | Iwi | 3 |
| Workshop | Industries | 8 |
| | Agencies | 4 |
| | Companies | 2 |
| Survey | Industries | 29 |
| | Agencies | 8 |
| | Communities | 3 |

## 4. Results

The research process and outcomes were documented here to illustrate the limitations of existing approaches to co-design that fail to take into account the mental models of a range of users in a system undergoing change. The way potential users relate to technology and the wider system, it sits within, can influence the efficacy of the technology in use and the adoption of intended systems change or improvement. We discussed the importance of revisiting assumptions about the existing biosecurity system in the process of sociotechnical innovation not only of diverse research stakeholders but also of the researchers.

As experienced biophysical and social science researchers working in biosecurity, we constructed two conceptual models of the sociotechnical system through interaction with stakeholders during the early stages of our research development. The first was a conceptual model of the biosecurity system that incorporates desired improvements, e.g., changes to social relationships (partnerships) and new technology innovations (mobile phone applications). In Geels and Schot's [52] terms, this model represents changes that put pressure on the current or incumbent biosecurity system. Jasanoff [21] might represent this as the sociotechnical imaginaries forming the aspirational goals or desired states of a collective future vision [53], e.g., those expressed by New Zealand's 10-year biosecurity strategy [39]. The second was a conceptual model of stakeholder engagement in the process of research inquiry that brings stakeholder perspectives into thinking about technology design. Rather than an aspirational future, this conceptual model is grounded in the context that participants bring to the process of research to produce results that are socially constructed and more likely to be adopted as they fit with users' mental models [54].

We considered these two conceptual models to be complementary and not fundamentally challenging or contradictory even though they direct research focus in different ways. From an epistemological perspective, one was a future-oriented realist and the other inquiry-based social constructionist. Their point of the intersection during research interactions would suffice for facilitating discussion in problem definition and solution-seeking. Such an intersection may be seen as providing a boundary critique for co-inquiry [44]. The engagement of stakeholders would support the process of both as a 'reality check' on our understanding of the incumbent or current system [52] and as a process of engaging in the social construction of its transformation [21].

*4.1. Conceptual Models*

4.1.1. A Description of the Future-Realist Model

At the conceptualisation stage of the project, the science team (the formal 'researchers' in this setting) circulated a potential model for general biosecurity surveillance in New Zealand to stakeholders to discuss their support for a funding application (Figure 1). This positioned the research positively in the eyes of the funding body that recognised direct connections with those intended to adopt the innovation. The concept placed those with responsibilities and/or vested interests in improving a public biosecurity surveillance system at the top of the diagram. These organisations included central and local government, primary sector producers and iwi who were in a position to collect data that could provide information about general surveillance needs. At the core of the conceptual model is the GSN that connects different stakeholders who communicate with their respective members about biosecurity surveillance needs (for example, the Māori biosecurity network, primary sectors, councils and schools). In this model, it was proposed that the ability of this stakeholder-led network to report observations of potential biosecurity threats would be supported by new technology solutions, such as mobile phone apps and other real-time communication methods that facilitated the transfer of observations to species identifiers.

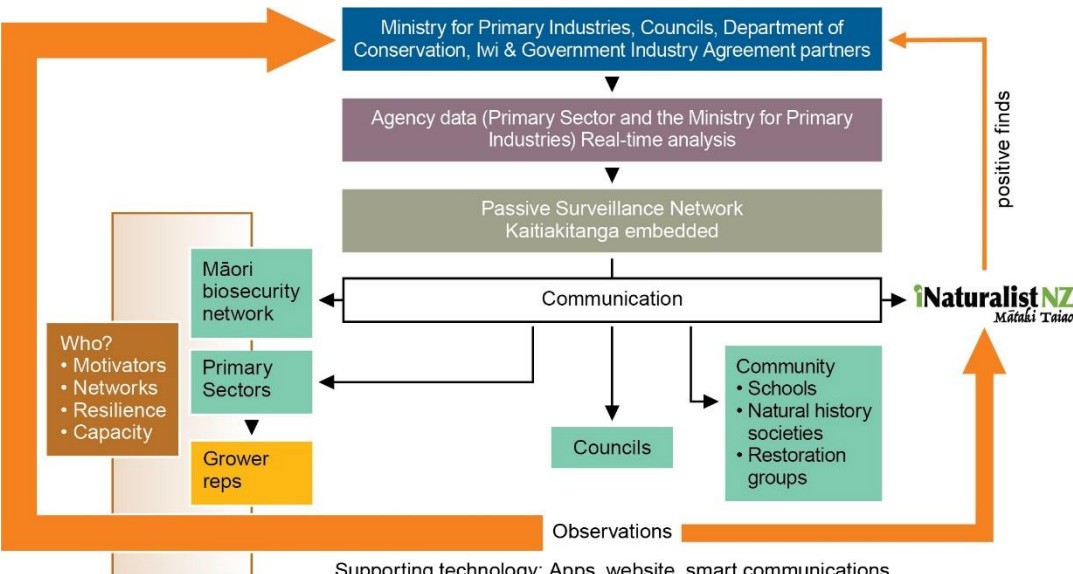

**Figure 1.** Conceptual model of a potential future general surveillance system circulated for consultation with stakeholders. Ministry for Primary Industries is the central government ministry that includes Biosecurity New Zealand.

At the outset, we were aware that Biosecurity New Zealand diagnosticians already managed a heavy workload. Hence, our conceptual model looked at two methods to accommodate this: (i) increasing the efficiency of processing individual observations by the use of smart digital technologies; (ii) by incorporating 'citizen scientists' (commonly perceived as data collectors, but in this case, also as species identifiers) to assist in screening low-risk observations that were less likely to be new to New Zealand with potential trade implications. Knowledge needs required to support the development of this concept were social, particularly relating to motivators (those who might motivate others/would be motivated to submit their localised finds) and networks (how people are connected through this practice of identifying biosecurity pests). We developed questions around how potential users/motivators and their networks could be harnessed to promote successful adoption of the proposed mobile pest identification tools and hence a stronger general surveillance system. Subsequently, the development of a second conceptual model articulating a process of stakeholder engagement in technology design was proposed (Figure 2).

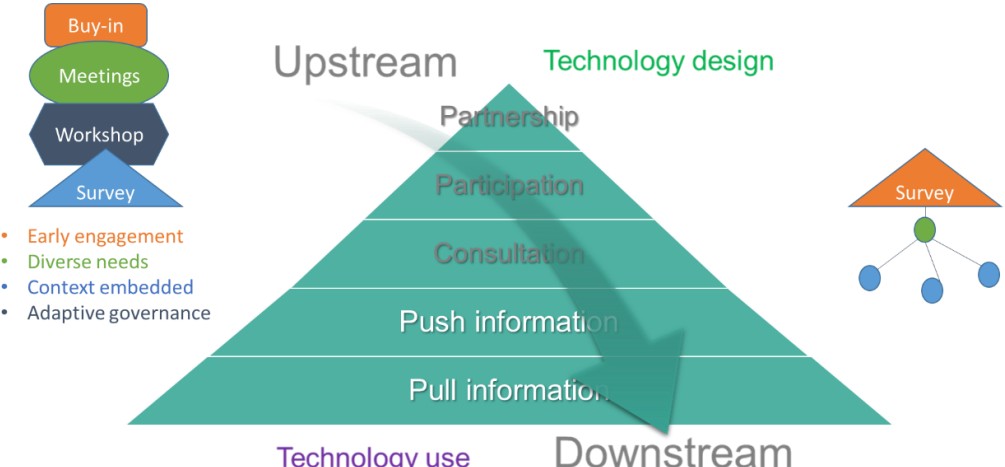

**Figure 2.** Stakeholder engagement conceptual model expressed in two parts. Overlaying upstream engagement in technology design with a model of stakeholder engagement [41,55], and the process followed for implementing the data collection and co-design framework (including buy-in, meetings, workshop and survey).

### 4.1.2. Stakeholder Engagement in Technology Design

A process followed for implementing the data collection and co-design framework with stakeholders was initiated (Figure 2), including (i) design considerations of stakeholders engaged during the buy-in phase of sharing our conceptual model (Figure 2); (ii) meetings held during planning phases of research; (iii) a multi-stakeholder design workshop; (iv) preparation of a survey instrument (a mix of open and closed-ended questions) with a sub-set of workshop participants that was then implemented by them and some of the other workshop participants. Four elements were elucidated by the first author during the early stages of engagement in project development meetings and emails to form an initial framework for engagement in technology design as (i) requiring early engagement, (ii) being contextually embedded, (iii) meeting diverse stakeholder needs and (iv) enabling adaptive governance. Adaptive governance, including the need to respond to uncertainty by promoting learning and encouraging constant monitoring of biosecurity outcomes [34], was something discussed by the researchers and realised at the design workshop as something that must emerge from the context of the STS innovation. Further buy-in to the innovation by iwi, other industry partners and agencies was perceived as a condition of successful implementation.

The benefits of the tool were considered to be the driver behind its uptake, across three layers of stakeholders. The first are the stakeholder organisations (industry, regional and local government and iwi); the second is with the individual submitters, and engaging their potential networks; the third is with the central authority in achieving early detection. There may be others, including the iNaturalist NZ species identifiers, where competitive behaviour regarding rapid and accurate identification is already evident. Review of such tools and their adoption in other settings revealed that incentivising use must come from the ground up to realise desired change [56]. Thus, the engagement of stakeholders in tool design was to support that ground-up process.

At the time of the writing this paper, governance of the surveillance platform was yet to be developed and a critical part of the emergent nature of the innovation within a changing context of who would resource biosecurity readiness activities via the government-industry partnerships (i.e., GIAs) in biosecurity and other potential partnerships between agencies and tiers of government. An additional player in this setting was Iwi, as the Treaty of Waitangi partners not previously engaged in biosecurity governance [8]. The national Māori biosecurity network (Te Tira Whakamātaki) was a key initiative that emerged independently of but during the process of the project. Te Tira Whakamātaki is a Māori researcher-led initiative aimed at connecting biosecurity governance structures and processes to iwi across the country rather than relying on a single group of Māori representatives to guide policy and its implementation [7].

A framework was initiated through building relationships and buy-in of our collaborators to focus on the outcomes of our processes of early engagement, looking at aspects of tool use in context and the diversity of needs for enhancing the biosecurity surveillance system. We found the need for feedback was an important missing element in the existing system, where people called into the '0800' number but did not receive any confirmation on whether they had reported a new to New Zealand species or not. A brief discussion based on our (i) workshop outcomes (platform design); (ii) existing networks (network extent); (iii) network survey (potential tool use) showed the need to balance complexity with ease of use, the existing limited reach of stakeholder networks and the importance of feedback and engagement for building trust. Together these are presented as three steps for developing an emerging surveillance platform. Following a presentation of the workshop and survey results, we discussed the implications of our evolving appreciation of stakeholder relationships, including tensions and role changes, and how they might be adaptively managed to support further investment in the governance of a GSN.

*4.2. General Surveillance Workshop Outcomes*

Having participants reflect on technology success/failures was a good way of encouraging critical engagement with the processes of design thinking. Participants expressed a desire for a solution that was simple, adaptable and reliable. However, they were also concerned that barriers to its use would need further investigation. Another critical step identified was clear communication channels between the design team, developers and users to ensure robust specifications that would meet expectations. It was important to participants that the observation tool or app development could be embedded within existing activities.

4.2.1. The Tool/Technology Platform

Participants did not have a clear single vision for an app as stand-alone or suite of tools that could be embedded within their existing apps or a website to manage and upload the submission of images and descriptions to a central system. Some even suggested these could be combined into one ICT system. However, the immediate constraints on being able to develop such a range of options were financial. At the very least, it was suggested, there needed to be a basic functionality that could be further developed in different contexts to meet a potential variety of surveillance stakeholders' needs and practices in different settings.

Participants also expressed a need for offline data capture as much of New Zealand, and especially the forestry sector, was out of current mobile phone range, and hence to maximise uptake, a standalone app was desired. There was also a preference from some to see the app embedded within an existing high-use app rather than adding to the existing competition between apps. A farm or forest management app was seen as a potential option. A third aspect was to be able to work with two-way functionality, which also fed into a fourth consideration, which was to encourage users to feel part of a community with information sharing and interactions between users.

Minimum data requirements: The basis of a discussion around data requirements was to ensure data input and options for reporting could meet the minimum conditions for early detection whilst also providing the potential to add further functionality as required. Keeping the technology to a simple, adaptable and flexible user interface meant that not too many fields for recording information should be included. Two aspects were considered—data inputs and reporting back to users about their finds. The essential fields considered for data inputs were:

- Observer contact details
- Date/Time
- Location
- Image

In the first instance, there was a need for reporting back on specific observations—notably, that reporting was time-sensitive and required immediate or semi-immediate feedback. Reporting observers would need to know if what they had found was new to New Zealand and if it was an introduced pest of concern to Biosecurity New Zealand. However, longer-range reporting of sector/regional performance was also a desired feature.

Specific features of these two reporting mechanisms were noted as:

1. *immediate reporting:* engagement with the user, acknowledgement of submission, timely identification, 'other' observations, BNZ follow-up if required;
2. *longer-range reporting:* performance reporting for a region or sector–context-sensitive to audience group/sector relevance, spatial location, regional/sector comparison, messaging/alert effectiveness.

In summary, there was a need to balance complexity with ease of use. Some questions were not essential for new-to-New Zealand pests and could form part of the other observations that might be sector/region-specific. Participants noted that reporting back needed to be sector-specific, and it was suggested that the science team produce a set of potential reports for circulation so that sectors could familiarise themselves with reports and how they might be used.

### 4.2.2. Exploring Existing Networks

Methods of identifying and producing a stakeholder analysis were tested through an exercise where participants mapped their stakeholder environment. This was a facilitated process during which some participants noted they had not done such an exercise before. Different lenses used to conceptualise stakeholders were suggested by the facilitation team, including mapping stakeholders by their levels of interest and influence or other conditions, such as levels of power or legitimacy, proximity or impact stakeholders [57–59]. Table 3 shows the stakeholder groups identified for each sector or agency as identification and reporting channels, along with potential tool users and possible triaging experts to check whether observations needed to be escalated to BNZ as a 'new to New Zealand' biosecurity alert.

**Table 3.** Identification and reporting channels, potential users and possible 'trusted advisors' to identify/triage observations by sector. We didn't have specific iwi participants to work with in this workshop as they were unable to attend on the day, and a separate process was followed through additional meetings.

| Sector/Agency | Identification and Reporting | Potential Tool Users | Possible 'Triaging' Observations |
|---|---|---|---|
| Forestry | specialist contract forest health observers, Forest Biosecurity Committee (FBC) members, some forest managers, NZ Farm Forestry Association interested members, scientists working in forestry | Industry via PineNet | scientists at Scion/Manaaki Whenua Landcare pathology/entomology, specialist contract forest health observers, Forest Biosecurity Committee members |
| Kiwifruit | growers, extension officers, contractors (some), orchard managers, technical advisors, Kiwi Wine Health (KVH) monitoring teams | farmers, contractors, farm consultants, intelligent advisors, influential adopters | technical staff in KVH/Zespri, Quality Assurance on post-harvest, Plant and Food research/Crown Research Institutes |
| Dairy | sector-specific biosecurity managers, some DairyNZ staff members | farmers, contractors, farm consultants, intelligent advisors, influential adopters | scientists AgResearch |
| Wine | sector-specific biosecurity managers | growers, vineyard managers, contractors, seasonal workers | |
| Apples and Pears NZ | sector-specific biosecurity managers | growers, orchard managers, contractors, seasonal workers, packhouses—quality | |

All participants recognised some limitations with their sectors/communities/organisations being able to identify plants, plant pests or plant diseases. For example, horticultural industries recognised that their seasonal workers came from varied cultural backgrounds that might require specific multilingual communication efforts.

There were also limitations within agencies. AsureQuality had national biosecurity capability but noted the need to have well-connected individuals with devices in the field. Regional Councils' knowledge and understanding varied from council to council. Department of Conservation (DOC) surveillance reporting typically followed a process of members of the public reporting to DOC rangers and then using in-house or external technical advisors available through email contact.

To explore some of these limitations, we asked participants to go back to their sectors, organisations or communities and engage with people who they thought might use the platform to ask them about their current practices.

*4.3. Post-Workshop Potential User Survey*

Following the workshop, a subset of participants worked through a set of questions that could be asked from potential users. The questions, developed through several iterations between researchers and participants to ensure relevance and accuracy, covered what tools potential users were currently using, who they shared information with and what feedback or information requirements they would like (see Table 1 survey outcomes, p. 14). While not all questions were related to the design of the tool per se, they did focus on the information inputs and role played by the user in informing others or sharing information. Participants thus demonstrated a mental model of how they currently applied an information network. A summary of responses to these questions helped with the design of the form, content and function of the tool.

A Generic Users' Model

A representative example of how the tool was being conceptualised by potential users (Figure 3) illustrates a generic model that the researchers constructed out of the workshop and follow on engagements through survey and design team meetings. However, at this stage of conceptualisation and development, although inclusive of stakeholders as potential users, the research could not reveal how people would incorporate this tool in their mental models of practice.

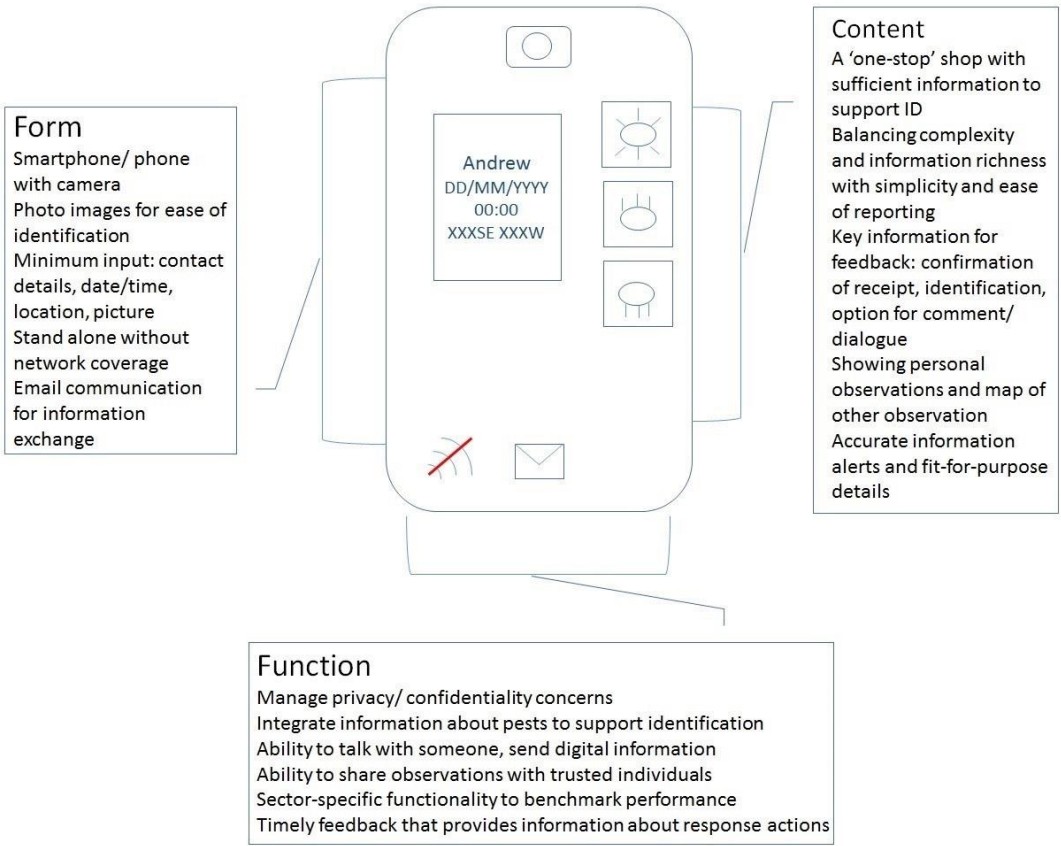

**Figure 3.** Information from users transferred to a generic model of form, content and function suitable as a basic set of conditions to meet users' needs.

The indications are that a tool to support citizen participation in general surveillance is desired and opportunities to share knowledge and receive timely feedback was important to potential users. Our proposed networking of such users with citizen experts as species identifiers from within a wider network of nature observers, such as iNaturalist NZ, could well serve this function. However, in spite of our efforts at achieving this level of confidence for buy-in on the ground, there were concerns from current system managers about the level of confidence in citizen experts, and the potential additional work pressure of increased citizen observers remained as issues that some were unwilling to accommodate.

### 4.4. A Rationale for the Constructionist Engagement Framework

A framework for the engagement of stakeholders and their networks was developed based on the power-sharing rationale of Arnstein's ladder of participation and a desire to move stakeholder engagement further 'upstream' to share perspectives in the ICT design process [2]. The idea that stakeholders would become a link in their networks of stakeholders, while assumed, can be made more tangible by having deliberate exchanges to support the development of knowledge sharing. Such a cooperative approach to working together with stakeholders was designed to ensure a 'safe space'

for experimenting with systems change [53,60] (p.711). Some of our stakeholders were leading the initiative by investing in and supporting its development while others were observing. The opportunity to bring together stakeholders as potential partners in the evolution of the STS (shown in Figure 2) was a critical part of engaging in collaborative means of developing solutions. An underlying rationale for doing this was to extend the reach of the research network [61]. In doing so, we were able to see more clearly what that networked looked like, gather some information on what different stakeholders' networks were currently doing in relation to biosecurity surveillance and to articulate what value they would place in the development of such as tool. Our co-design thus revealed different mental models of potential users and drew upon common threads, whilst also identifying knowledge needs beyond the general purpose of reporting biosecurity observations (Appendix A). The resulting design specifications provided a skeleton of basic needs that could be adapted to different stakeholder skins (Figure 4), meeting diverse purposes as encapsulated by the engagement framework.

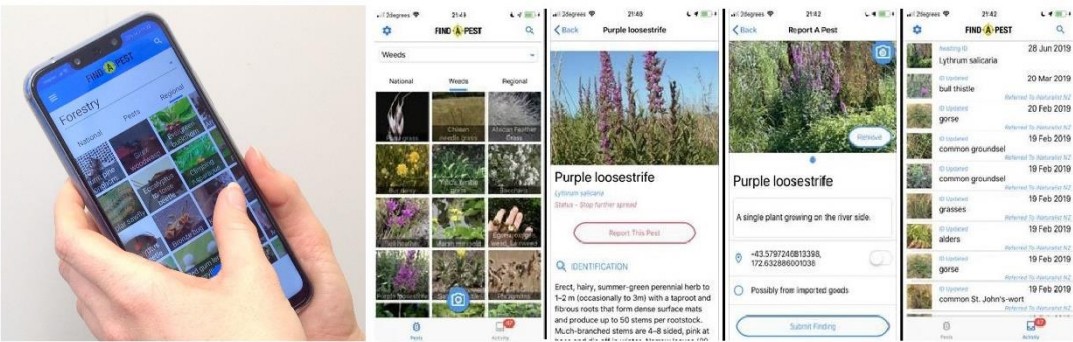

**Figure 4.** The resultant model in use for forestry and some screenshots from the Find-A-Pest app, including gallery, identification, submission and activity screens for regional weeds.

## 5. Discussion

The strength of using this conceptual model and approach for extending technology design engagement through stakeholder networks was connecting a diverse group of potential immediate users from different agencies, sectors and communities. It did not capture those outside of these networks, nor did it consider whether other stakeholders might reveal different needs for content, form or function. Nevertheless, the exchange with participants enabled a set of criteria that could shape an understanding of the development of the platform to enable a practical tool to service expressed needs. It is also evident that potential users provided a set of guides that went beyond the minimal requirements of data inputs. For example, survey respondents identified potential people they would share information with as well as a set of feedbacks and engagements they would require to connect with a trusted set of users. They desired to use the tool for educational purposes as well as a performance tool to measure reporting against effective responses, e.g., controlling outbreaks.

### 5.1. Research Reflections

By engaging with both future-oriented realism and inquiry-oriented constructivism reasoning processes, we have found a means to align our different epistemologies and conceptual models and re-think our characterisation and engagement of stakeholders from the bottom-up of sociotechnical innovation. Such a development has been important as we better understand the innovation of our systems that leads to more participatory approaches to biosecurity surveillance to complement government efforts. This understanding has been based on learnings from the practice of our collaborative inquiry that enabled an opportunity for us to reconsider original conceptual models and their weaknesses relative to the problems encountered. Our two model approach provided a basis for connecting to the potential future reality, and, at the same time learning from our engagements about the limitations of that model in the present. It is the conflict in the rationale that enabled us to

reconceptualise. Rather than seeing the end-users as externalities to the current system, there was a clear gap in our model that did not incorporate the other side of the system of users involved in general surveillance; namely, the recipients of public biosecurity submissions through the 0800 number as *official diagnosticians*. We needed to address the agency of stakeholders in different parts of the existing biosecurity surveillance system to adequately move towards a regime shift in STIS terms [27], and where supported by landscape changes in the operational environment such as: i) recognising the importance of early detection for minimising costs and impacts; as well as ii) willingness of citizens as *observation submitters* to use mobile ICTs to support general surveillance. These alignments were critical to effectively invite transformation through our inquiry processes but also in a way that emancipates users and their values for more efficient and effective biosecurity responses through early detection [2–4]. The final factor in our system change is the *species identifiers* [62], still a critical component of the innovation that has not yet been investigated from a social research perspective. Further investigation would be needed to see whether the skill and capability of this network of identifiers would grow through this platform or become overburdened by the increased demand for their expertise that is given voluntarily.

*5.2. Implications*

Our design considerations included the development of a framework in which early engagement could support the creation of a tool that would meet end-users' needs. We also recognised the need to meet diverse levels of interest and influence amongst participating stakeholders, and to ensure that the platform could be embedded in different contexts relative to the needs of different users or stakeholder networks. Finally, we acknowledged the need to create an adaptive governance model to support the evolution of stakeholder needs and tool use (this has been one of the more challenging aspects and is ongoing). However, moving engagement in technology design upstream has created new kinds of consideration that reveal points of friction within the existing biosecurity system and its relationships that need to be managed. As with Zhu et al. [63], the costs of change are an important consideration for users in the system where 'excess inertia' in sociotechnical systems (STS) can make it difficult to shift to new standards of practice. The need to realise the co-design principles of our action research framework—early engagement, diverse needs, context embedded and adaptive governance—across a range of settings from the centre to the periphery of the system undergoing change requires further consideration. Time is needed to support an embedding of desirable and feasible change in practice with a wider range of operational actors in this sociotechnical system innovation. We had been operating within a bottom-up framework and now must turn to the effects of that bottom-up change on top-down operations. A need remains to create a nomenclature for different kinds of actors in this sociotechnical innovation system (STIS) and to appreciate different logistic and political imperatives of the actors as differently situated end-users to better understand motivations to engage in/resist systems change.

Geels [27] noted the limitations of STIS in being able to account for the agency of actors involved in the system. Despite being the most frequently used framework in research projects exploring and analysing sociotechnical change, Sovacool and Hess [53] illustrated the strength and weaknesses of STIS compared with other approaches. They noted the importance of understanding the mental models of people in a complex system and how they can clash with or contradict the operational logic of the overall system and how it performs. According to Geels and colleagues [27,52,64], niche innovations need to be sufficiently developed to enable a change at the level of regime. Here, we have only been dealing with one end of the change in our considerations of upstream engagement (that with the users at the periphery of an ICT mobile surveillance system) and in which there are other innovation niches requiring attention (e.g., with those receiving submissions and impacted by their quality or quantity). Shared cognitive routines [52,65] across different settings of change need to be reflected upon. As noted by Geels and Schot [52], "scientists, policymakers, users and special interest groups also contribute to the patterning of technology development" [66] (p. 400). Within the existing biosecurity

system, there are also actors (e.g., within agencies and sectors) that present barriers to system change because of concerns about adverse impacts on the validity and performance of the current system, institutionalised in trade relations through the WTO Sanitary and Phytosanitary (SPS) Agreement that are only partially equipped for adapting to change (see [34]). The risk of having an unofficially early detected new to New Zealand invasive species that could close export markets still carries weight. The wider effect of failing to detect early is less appreciated. As noted by Sovacool and Hess [53], there is a need for a clear definition of the level you are operating [67] (p. 54), as niche innovation, regime or landscape and aligning interactions between them to support rather than disrupt the transition [52]. We have addressed a point in which there is an opportunity to reflect and realign to the emergence of new stakeholders in the STIS not previously engaged in the research (*official diagnosticians* at the centre of information feed) or the biosecurity system (*specifies identifiers* in the pathway of information flow). However, due to a rhetorical repositioning (from the importance of deficit in general surveillance to the importance of systems change), this also requires a shift in our original conceptualisation of the problem.

## 6. Conclusions

This research aimed to engage stakeholders in the improvement and enhancement of the existing biosecurity general surveillance system in New Zealand. It contributes to discussions about the development of forest and tree health biosecurity through the initiative of the forestry and horticulture sectors co-investing in sociotechnical innovation of general surveillance. By drawing upon new ITC developments of smartphone and imagining capability together with the recruitment of citizen scientists, not just as observation submitters but as species identifiers, it proposed to increase capability for early detection of invasive pest species. Subsequently, new challenges arose in increasing participation in the biosecurity system, from addressing a surveillance deficit to coming to terms with systems change.

From the outset, we wanted to think about project design—how that can be influenced by the rhetoric of project development and cultural context—including how 'designers' (the researchers and stakeholders drawn into the design process) influence project framing [23]. It was also our intention to introduce a method for considering a more dynamic environment of STS change and transition [68] in response to the emergent nature of stakeholder relationships. There is a need to be responsive to the existing needs of stakeholders and potential technology users within the existing system as well as attend to the need for developing a community of users that gain benefit from their interactions as part of the emerging surveillance network. Thus, we recommend designers incorporate time to reflect on project conceptualisation as part of the innovation development process; something that allows for iterative and recursive thinking about project conceptualisation in the light of new understanding [22]. Here, we have reflected on the project framing and conceptual models—and reconfigured our project development. In part, this has been achieved through the inclusion of applied social science—to be able to shift from deductive to inductive frames with our biophysical scientists as STIS co-researchers. Opening up thinking about systems change dimensions that were present in the project conceptualisation but not made explicit through the research methods, by recognising users at the centre of the surveillance system, was a necessary step for realising this development. As noted by Mulder [69], acknowledging the change to a stable system or one, whose stability is socially constructed, can mean stronger resistance to the innovation than a technical solution can address. Thus, ensuring that the social needs and supports for the transformation will not undermine the current system is as important as ensuring that the technology works and will be used.

In this research, we were able to introduce social science perspective as a gradual transition and find opportune moments to reflect on project design. However, along with others, we concluded that this needs to be built into sociotechnical design expectations (through agility, flexibility and emergence) to better manage projects and the learning and surprises they encounter [23,24,70]. Our finding was consistent with other sociotechnical innovation theory and knowledge developments that acknowledge the importance of agency or deliberation over change implications [27,71]. For example, Lancaster

and Yeats [72] suggested the need, when working across a larger array of stakeholders within an operational setting, to use scenarios and problems as a course for understanding different perspectives and bringing them into the design process as part of problem-solving. Criticisms can be levelled, and internal constraints can be identified [72] through inclusive design processes. As the usability of new tools or set of tools is experienced across a network, the practices of identifying, reporting and learning can surface new shared meanings or values, which our co-designers found to be an important emergent dimension of building a community of users. However, we found that changes in stakeholder relations in a more lateral system of surveillance need to consider the mental models and learning needs of users at the centre of an information system feed and not just its periphery where information is fed in. The socio-political structures of information control requiring biosecurity agencies to service WTO requirements and trade relationships may be a limiting factor for supporting more open forms of biosecurity through which new information processing systems can be incorporated into a general surveillance network.

**Author Contributions:** S.M.P. led the project and initiated the concept of an improved surveillance network using mobile ICT, gaining funding and the commitment of key stakeholders. S.M.P. and A.G. met with stakeholders in the early meetings and ran the workshop; they jointly developed the survey questions with a subset of workshop participants and analysed the results to provide details on the form, function and content of the app. A.G. developed the social construction methodology to complement to the future realist proposal, including reviewing the literature on sociotechnical systems and their innovation, and drafted the original paper. M.M. supported the conceptual development of the sociotechnical information system analysis and provided social science guidance for the paper, reviewing earlier drafts and making recommendations. S.M.P. contributed to the introduction and background for the paper, particularly for an understanding of the New Zealand biosecurity and surveillance system, and how it could be improved, and reviewed earlier versions of the paper with some technical and conceptual critique.

**Funding:** This research was funded by the Ministry of Business, Innovation and Employment (New Zealand's Biological Heritage National Science Challenge, C09 × 1501, project BH2.5 A passive surveillance networks protects New Zealand's biological heritage from biosecurity threats), Envirolink Tools (sponsored by Environment Southland), Biosecurity New Zealand, the New Zealand Forest Owners Association and Kiwifruit Vine Health/Zespri.

**Acknowledgments:** The authors would like to thank the New Zealand Bio-Recording Network Trust who facilitate the iNaturalist NZ community and additional agencies for their participation in the co-design workshop and stakeholder engagement: Apples and Pears NZ, Avocados NZ, AsureQuality, Auckland Council, Biosecurity New Zealand, Department of Conservation, Dairy NZ, Hawke's Bay Regional Council, Horticulture NZ, NZ Wine, Te Tira Whakamātaki, and Wakatū Incorporation. Special thanks to Brendan Gould, Lyndsey Earl and Jo Lloyd of the Biosecurity New Zealand Surveillance and Incursion Investigation team, Randall Milne from Environment Southland for their continuing support of the project and Will Allen of Will Allen and Associates who facilitated the workshop and provided action research guidance.

**Conflicts of Interest:** The second author is a trustee of the New Zealand Bio-Recording Network Trust. Those contributing funds to the research programme also contributed to the co-design along with other stakeholders, as well as participated in collecting data for the research, facilitating the development of an appropriate tool that could be used and validated by user communities.

## Appendix A

Sample mental models generated from data collected through co-designed stakeholder surveys of stakeholders as potential 'end-users' from the community, agency or industry networks of workshop participants.

**Community A**

**Action taken currently**

Photograph them

**Technology advice (used to understand)**

Cell phone camera because photo comes tagged with date, time, location. Apps that provide identification assistance, if I have coverage

**Method of reporting (preferred methods)**

If I thought it was a species of interest to [BNZ], I would prefer to photograph it using my cell phone, and send it to an [BNZ] email address, or an [BNZ] website. I don't want to waste my time talking to someone who is not an expert on the [BNZ] hotline. [BNZ] can get in touch with me if they want to follow up the photos. If I had no reason to believe it was a species of interest to [BNZ], then I would use [iNaturalist NZ] as a first call. If on identification it turned out to be a species of interest to [BNZ], then I would send a link to the record to [BNZ] (preferably by email or website or app). I would only think of reporting to [BNZ] if I had strong suspicion that it was an organism I had seen in an [BNZ] advert. In most instances, I would assume it was an exotic resident or unknown native. [iNaturalist NZ] is a better use of my time (completed in 2-4 mins), and more rewarding than ringing the [BNZ] hotline (generates an ongoing conversation with both professionals and amateurs, creates a long term national record, I get useful interactions and networking).

**Current info sharing method**

[iNaturalist NZ] – it's designed for the task of crowd sourcing identifications and keeping long term national records. Need expert opinion before I gather samples for a museum or a crown research institute. Can't pick, press or pickle every unfamiliar thing I see.

**Capture and advice technology**

Smartphone and iNaturalist [NZ] app (formerly NatureWatch)

**Usefulness of current tools (what could make them easier)**

Very useful and makes me part of a community of nature watchers

**Community/sector advice (sharing or seeking)**

Other people on [iNaturalist NZ], DOC colleagues, subject matter experts

**Desired feedback**

Species identification, status, distribution, if invasive then extent and type of impacts in other countries, thanks.

**Desired dissemination**

My sector is natural heritage conservation. I would like to see reports of suspicious or unknown organisms on a shared public platform, so that information reported to [BNZ] is visible to the public, with the proviso that the person reporting can conceal their identity or conceal the location from the public (but not the database administrators). It's called transparency and accountability.

**Information sharing support**

Use of [iNaturalist NZ] for this task

**Barriers to reporting**

Being too busy. Being unconfident about the significance of the sighting. Having low expectations regarding [BNZ] willingness to act, that is: expecting [BNZ] to decide that attempting eradication is too expensive, that costs outweigh benefits, or that funds should come from somewhere other than government. [BNZ] needs to create incentives for reporting because reporting costs personal time and comes without personal benefit, eg an incentive might be that a report equals a vote for an environmental charity to receive funds. [BNZ] can pick a set of environmental charities that can receive votes

**Overcoming constraints (if felt uncomfortable)**

A rewarding encounter eg access to an engaged expert who is looking at the photos I took as we speak.

**Desired feedback (confidence in handling information)**

If it turns out to be a border incursion, gratitude/recognition, plus feedback on what actions [BNZ] has undertaken or will undertake.

**Figure A1.** Sample mental model 1.

| Community B |
| --- |
| **Action taken currently** |
| N/A |
| **Technology advice (used to understand)** |
| Google/Internet. Look at government organisations websites ([BNZ] /DOC/Auckland Council) |
| **Method of reporting (preferred methods)** |
| Email and/or text. |
| **Current info sharing method** |
| Email. |
| **Capture and advice technology** |
| I don't have specific apps for this and think if there was it is a limitation as some people may not want to have to download the app. App is one option but having alternatives is a good idea – email/text/online form/app. |
| **Usefulness of current tools (what could make them easier)** |
| I mostly use email for any reporting issue. I wouldn't be inclined to download an app specifically |
| **Community/sector advice (sharing or seeking)** |
| [BNZ] /Auckland Council |
| **Desired feedback** |
| Location, time, date, photo upload function, general description, and contact details to be contacted at for further queries. |
| **Desired dissemination (in sector/industry)** |
| Email. |
| **Information sharing support** |
| Ease of forwarding on. Make the message simple and if people want detail have a link for them to go to that. Everyone is time poor and not likely to read too much detail in initial emails |
| **Barriers to reporting** |
| Privacy of my details. Lack of certainty it is of interest |
| **Overcoming constraints (if felt uncomfortable)** |
| Some sort of pre-reporting checklist to see if what you are reporting may be something the organisation is interested. No one wants to look stupid. |
| **Desired feedback (confidence in handling information)** |
| Some sort of acknowledgement on how the individual's personal information would be stored/used/referred to other parties. |

**Figure A2.** Sample mental model 2.

| Industry A |
| --- |
| **Action taken currently** |
| On two occasions over 20 years i have been concerned about a clematis vine in our trees. I removed samples of it and discussed them with a [company] rep and also looked on the internet for photos. I identified the vine as a native clematis. I have never been aware of observing any unusual insects in our forest |
| **Technology advice (used to understand)** |
| I would probably search the web and also seek guidance from [company] should a problem arise.) |
| **Method of reporting (preferred methods)** |
| I would initially contact our [company] rep and follow his guidance. If I deemed it to be extremely urgent, I would look for an [BNZ] "hot line" telephone number or an [BNZ] e mail address if I could find one. |
| **Current info sharing method** |
| I have never found any in our forest but on two occasions over the years, I telephoned [research institute] and in later years [research institute], to discuss unusual insects I identified on out [region] property. |
| **Capture and advice technology** |
| I would use a conventional camera to take photos. I have a mobile telephone BUT it is not a smart phone. I will eventually upgrade to smart phone technology. However, I have ready access to a desk top computer when at home. |
| **Usefulness of current tools (what could make them easier)** |
| I find my desk top computer to be very useful. I guess a smart phone would give similar coverage when away from home (but there is no cell phone reception at our forest so any identification apps could not be accessed until the outskirts of [township] were reached). |
| **Community/sector advice (sharing or seeking)** |
| [Company] rep |
| **Desired feedback** |
| I would hope that [BNZ] (or an appropriate agency) would acknowledge the registration of the observation promptly and give appropriate advice. |
| **Desired dissemination (in sector/industry)** |
| As a member of the [industry association], I am sure our organisation would be informed and they would in turn e mail us promptly if a potential problem was to arise. The [industry association] works very well in this respect. I am not an expert in communications, but I imagine main stream news outlets (radio, TV and newspapers) may reach many members of the public. From memory the District Council may have records of commercial forests in the region. |
| **Information sharing support** |
| Continue to be a member of an organisation such as the [industry association] |
| **Barriers to reporting** |
| I would like to think that I would report any problem very promptly (and then hope that it was a false alarm) |
| **Overcoming constraints (if felt uncomfortable)** |
| N/R. |
| **Desired feedback (confidence in handling information)** |
| I believe it is very important that any person who reports a potential problem is treated with respect and is taken seriously, no matter how trivial the reported incident may turn out to be. Most of us are not experts in such matters and a prompt and courteous response is likely to encourage ongoing participation with such an important issue... |

**Figure A3.** Sample mental model 3.

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
