# Peer review of "Emerging Stakeholder Relations in Participatory ICT Design: Renegotiating the Boundaries of Sociotechnical Innovation in Forest Biosecurity Surveillance"

_forests, doi:10.3390/f10100836_

Round 1
Reviewer 1 Report
This article discusses the integration of a number of stakeholders and initiatives for improving detection strategies in New Zealand. While in my opinion this is not a scientific article per se, it is important that such discussion are completed and presented so that the scientific research completed on detection programs can be compared and integrated within the technological and socioeconomic frameworks of the overall monitoring strategy.
I fell that it is overly wordy and verbose. This is a relatively minor comment, and something that cannot be pinned down to specific sections; however, any final edits of this paper must seek to ensure that brevity to the manuscript be completed. This is essential for the reader to understand the content and directions the manuscript is suggesting.
Author Response
Reviewer one appears to have accepted the article without specifically indicating that it can or must be improved. Whilst they recognise that it is not a scientific article they do state the importance of the discussion for comparing detection programs and their integration within “technological and socioeconomic frameworks of the overall monitoring strategy”. We welcome this comment as an indication that such social background to research is a valuable part of understanding the context of detection programs and associated scientific inquiry.
However, they do suggest that “any final edits … must seek to ensure brevity” to ensure the reader can be clear on “the content and directions” of the manuscript. We did not feel that there was a need to modify the manuscript based on this comment but to ensure that any further modifications were kept brief and to the point.
Reviewer 2 Report
In "Introduction", there are some subheader-like lines (lines 38, 60 and 87) without any numbers, bold or italic. Also, What is the figure on line 36? Are these allowed on the journal format?
Authors should compare the situation between New Zealand and other countries more. Actually in UK and USA, the participatory ICT have been adopted for invasive alien species.
I feel the use of iNatulalist is not good for, in particular, Forest biosecurity survaillance, because it do not work well for species identifications of tree pathogens. The subject of this ms was on the stakeholders relations and this comment may be beyond the subject but it is also true that some difficulties still exist on the ICT for biosecurity survaillance.
Author Response
Reviewer two indicates that the introduction can be improved, noting that the sub-header-like lines (ll. 36, 60 and 87) and the figure on line 36 be adjusted. The figure is a graphical abstract as an optional addition to the textual abstract, which we have removed from the manuscript and now added as a separate file. The unidentified sub-headers are important markers for the reader to guide them through the three parts of the introduction. These have now been identified as sub-headers and an additional higher-level header created for a background section (l. 128) following the introduction. An additional sentence has been included to refer to use of participatory ICT in other countries (ll. 71-77), and to note comparisons with the development of participatory ICT for general biosecurity surveillance in New Zealand (ll. 83-85).
Furthermore, as noted by Reviewer two, we acknowledge the limitations of the work to date as not being inclusive of plant pathogens but have not altered the paper, as they suggested, this was beyond scope. Pathogens are largely excluded from the platform development as many pathogens are difficult to distinguish by visual symptoms. At this stage we have only included pathogens that are visually distinctive, e.g., myrtle rust, in the platform. One of our intentions is to use the platform as a basis for developing knowledge and we hope to increase the number of pathogens included over time, as users may develop more knowledge into the future.
Reviewer 3 Report
This paper presents a research outcomes on stakeholder relations in a biosecurity sociotechnical system (STS). The methodological concept and research motivations are well addressed. It is interesting to see how the advanced IT can be linked to enhancement of biosecurity in forest management.
I think the basic message in the paper is clear and here is comments for minor revision below.
My understanding is that the focus of the paper is the relationships within operational environment in STS. Nevertheless, it is also quite important to identify incentives mechanism within STS to construct a robust system. The paper needs to address a matrix of incentives for stakeholders. Section 3 discusses about interests of potential partners a little bit, but it is not sufficient to convince readers. This paper is about advanced technological applications to forest surveillance system. It is necessary to present a concept model for the employed technology: e.g., image contents the verification methods on them, type of API if applied. Is there any possibility that this STS can be associated with, e.g., local currency so that participants can get financial incentives or rewards? Image files needs to be presented in the paper to show outcomes of field works. There are currently only concept images on this project.
Author Response
The third reviewer, has indicated that the research design could be improved with minor suggested revisions. Although noted that our paper is about relationships within an operational environment – they would like us to identify ‘incentives mechanism’ within a sociotechnical system to enable its robustness. We would like to acknowledge that this paper reports on the design process and not the incentives to use the technologies. In principle, we refer to the engagement of stakeholders and potential users in design as a mechanism to support design that is relevant and user friendly, thus more likely to be adopted. Based on our research, there is evidence to suggest that financial incentives do not work for this kind of development and that it also may develop perverse outcomes, overburdening the resources of species identifiers. This might be something to look at in more depth in the future.
However, we do note that the government is promoting New Zealanders generally to become involved in biosecurity as part of their strategy 2025 (see ll. 169-172, 187-189), and that this tool anticipates greater public usage – as well as the need to manage the inflow of reports from a more active citizenry. For this to occur, we rely upon the stakeholders support ‘in-principle’ of the tool as one of the means for citizens getting involved in New Zealand’s biosecurity system. However, to re-emphasise our main argument, we are not just focused on the incentivization of the tool (although this is an important goal), but on how it will impact on the existing system and its resources, e.g., responding to a higher number of observation submissions.
Nevertheless, we appreciate that we may have not made the existing incentives clear enough for the reader and therefore have added an extra sentence to Section 3 (now 4), outlining the benefits of the tools across a number of user sites (ll. 364-371). The first is the stakeholder organisations (industry, regional and local government and iwi); the second is with the individual submitters (and engaging their potential networks) and the third is with central authority in achieving early detection. There may be others including the iNaturalist NZ species identifiers, whereby competitive behaviour regarding rapid and accurate identification, e.g., of weeds, is already evident. Nevertheless, we acknowledge that we are currently unsure as to whether this network of identifiers will grow through this platform or become overburdened by increase in use. We have added an additional sentence to highlight this point (ll. 555-557).
In their reference to ‘field works’ we are not sure whether the reviewer refers to outcomes of stakeholder engagement fieldwork or of the app in use. Thus we have provided images of the app in use, and of the identification and reporting pages (screenshots) of the app (ll. 518-519). We have already provided examples of the survey results in the form of ‘mental models’ of potential users in the Appendix A.